# Influence of Sports Participation on the Behaviors of Customers of Sports Services: Linear and Qualitative Comparative Analysis Models

**DOI:** 10.3390/healthcare11091320

**Published:** 2023-05-04

**Authors:** Fernando García-Pascual, Ignacio Ballester-Esteve, Ferran Calabuig

**Affiliations:** 1Departament d’Educació Física i Esportiva, Universitat de València, 46010 Valencia, Spain; ferran.calabuig@uv.es; 2Department of Teaching and Learning of Physical, Plastic and Musical Education, University Catholic of Valencia, 46110 Godella, Spain; ignacio.ballester@ucv.es

**Keywords:** sports frequency, fitness center, sport management, behaviors, healthy life, satisfaction

## Abstract

Sports services have become an important enclave for our society. Due to its complete sports offer, customers can perform physical activity that helps them to improve their health and well-being. In fitness centers, managers try to know what their customers are looking for in order to create more personalized experiences, as well as to improve their health. This study seeks to understand the influence that sports frequency has on the future behavior of users of a sports center, using two complementary methodologies. A sample of 383 users of a private sports center was used. The two complementary methodologies used were linear models and comparative qualitative analysis, based on the combination of sets. The results show how sports frequency influences the process of creating users’ future behaviors. Considering that none of the variables are necessary, it is observed that perceived value has a significant influence on users’ future behaviors. The use of two complementary methodologies provides a more complete understanding, which helps sports managers to plan and manage effectively to ensure user satisfaction and loyalty. In addition, facilities can incentivize customers through loyalty programs and promotions to maintain their engagement, as well as healthy styles to encourage service recommendations.

## 1. Introduction

Physical activity is an important part of our lifestyle as it has multiple physical and mental benefits. However, despite its obvious benefits, current physical activity in the population is insufficient. According to the World Health Organisation’s Physical Activity Report, only 23% of adults worldwide meet the minimum recommendations for daily physical activity (150 min of moderate activity or 75 min of vigorous activity per week). Moreover, sedentary lifestyles are a growing problem, with an average of 3 h a day spent sitting for adults and 8 h a day for students.

One of the main causes of sedentary lifestyles is lack of time and lack of access to safe and suitable places to exercise. In addition, technology and sedentary life at work are also important factors in the decrease in physical activity. Even so, for some time now, a change in the sporting culture of citizens has been observed, gradually increasing their active participation. This change may be due to an increased awareness of the importance of health or increased accessibility to sports services, such as sports centers. These sports centers have become safe and suitable places to practice sports nowadays, as they are designed and equipped to facilitate the practice of different activities. This allows people to choose the activity that they like the most and that suits their needs and goals. In addition, many sports centers offer group classes and personal trainers who can guide and coach athletes to achieve the best results.

Recent research has highlighted the importance of understanding the reasons why people join and continue to participate in sports programs. However, a large number of those who start a sports program or enroll in a sports facility drop out within a year [1,2].

Another advantage of playing sports in sports centers is the atmosphere and company that can be found in these places. Many people enjoy the sense of community and belonging that comes from sharing a space with others who have similar interests. In addition, doing sports in a group can be more motivating and fun, as experiences and challenges can be shared with other athletes.

Different research has long analyzed the behavior of the customers of these sports services [3,4], with the aim of finding out which aspects strengthen the loyalty of the users, in order to allow lasting viability of the service. Therefore, studies have addressed variables such as customer satisfaction, perceived value, and psychological aspects such as emotions, in order to have very detailed information on user perceptions. Different models have been defined, in which through the connections of the variables, researchers offer the managers of these sports services results that they must convert into tools that allow them to identify areas of improvement in the service. Through the analyzed perceptions, managers of these services can identify trends in the needs of users, allowing them to make more informed decisions, which in turn will allow them to have a high degree of success.

Therefore, the aim of this study is to determine the influence of the frequency of participation and attendance to supervised classes by users of a sports center on their loyalty to the sports service through two different methodologies.

## 2. Theoretical Framework

### 2.1. Customer Perceptions of Fitness Centers

One of the main advantages of sports centers is their ability to provide access to a wide range of exercise equipment and facilities. This is particularly important given the limited opportunities available for physical activity in modern society. Sports centers provide a convenient alternative that offers a safe and controlled environment for exercise [5]. Within the complex general model of sports management, one of the most analyzed indicators is the perception and behavior of users. Indicators such as satisfaction, perceived value, or loyalty have been analyzed in the literature [6,7]. In this context, fitness centers play a very important role in the provision of these sports services. Knowing this perception of the consumers of physical activity has become a fundamental tool for these sports centers to understand their future behavior. Within the specific literature on sports management, there are many studies that have analyzed these perceptions after the interaction of users with the sports service [7,8,9]. In these sports services, the loyalty of their users plays a fundamental role in order to guarantee their viability. This loyalty, understood as a behavior, is closely related to user perceptions that occur in the interaction between the company and the user. Customer loyalty has been a fundamental factor in the study of user behavior in sports centers, since obtaining positive levels of this variable is proportional to guaranteeing the viability of the service. Within the field that analyses the behaviors of these users, there have been multiple studies that have analyzed this attitudinal variable [6,10,11,12]. Tsai et al. [13] in their work analyzed this variable, concluding that when users obtain high levels of loyalty, they will be more likely to recommend the service or to encourage positive word-of-mouth.

Within the literature on the management of sports centers, there are two variables with a significant relationship with these future behaviors: satisfaction and perceived value. Vieira [14] finds significant relationships between perceived value, satisfaction, engagement, and behavioral intentions. In their work, Moura e Sá & Cunha [15] conclude that users who are satisfied with their experience are more likely to return and recommend the pool to others, indicating the importance of these factors in building customer loyalty. Furthermore, Gjestvang et al. [16] conclude that understanding the reasons and barriers to exercise adherence is important for developing effective strategies to promote sustained exercise behavior.

### 2.2. Frequency of Participation of Fitness Center Customers

As the years have gone by, new, healthier lifestyle habits have been acquired, such as an increase in the practice of sports. Nowadays, it has become normalized to practice physical activity on a weekly basis, with a large part of society, across a wide spectrum, doing physical activity both outdoors and indoors, such as in sports centers. This increase in sports practice is also reflected in sports services such as fitness centers, where according to the European Health and Fitness Market Report 2022 [17], in Spain at the end of 2021, the total number of users of sports centers stood at 4.8 million, suffering a slight increase compared to 2020, where 4.3 million customers were counted.

Sports frequency is an important factor for those seeking to improve their performance in endurance sports, such as cycling and athletics. According to Barker et al. [18], increasing training frequency can improve performance. However, Bosquet et al. [19] point out that there is also a tipping point where increasing sports frequency can lead to overload and result in decreased performance.

Age is also a factor to consider in appropriate sports frequency. According to Katzmarzyk et al. [20], as we age, our body tends to lose muscle mass and increase body fat, which can affect our ability to perform physical activity. Therefore, it is important to adapt sports frequency to age and individual fitness to obtain optimal health and performance benefits. Sports frequency may also vary according to the type of physical activity performed. According to the American College of Sports Medicine (ACSM) [21], resistance training, such as weight lifting or flexibility exercises, is recommended two to three times a week, while cardiovascular training, such as brisk walking or running, is recommended three to four times a week. However, it is important to note that not all sports activities are equally beneficial to health. According to the study by Smith et al. [22], some sporting activities, such as running or cycling, have a greater impact on cardiovascular health than others, such as yoga or Pilates.

Sports frequency in sports centers is essential for improving people’s health and well-being. Sports centers offer a variety of exercise and activity options to meet the needs and preferences of a wide range of people. It is important to continue to promote and encourage physical activity in sports facilities to improve the health and well-being of the general population.

### 2.3. Guided Activities as a Sports Offer in Fitness Centers

According to Masuki et al. [23], the lack of control over the characteristics and types of activities in a fitness center may affect adherence and make it difficult to identify the determinants of sports practice. However, fitness classes in gyms offer a structured and guided training experience for those seeking to improve their fitness.

One of the main benefits of gym-based fitness classes is the opportunity to participate in structured and supervised exercise. According to a study by Smith et al. [22], participation in structured exercise programs has been shown to improve fitness and overall health outcomes. Specifically, the authors found that individuals who participated in fitness classes experienced improvements in cardiovascular endurance, muscular strength, and flexibility.

In addition, many fitness classes are designed to provide a high-intensity interval training (HIIT) experience, as noted in Gibala et al. [24], which has been shown to be effective in improving cardiovascular fitness and increasing metabolism. In addition, the use of equipment, such as weights or resistance bands, can provide a greater challenge and help build strength and muscle mass.

A study by Rokka et al. [25], found that individuals who participated in group fitness classes at a fitness center experienced decreased stress levels and improved mood. On the other hand, Baena-Arroyo et al. [26], in their work on the comparison of professionally or virtually led classes in the context of fitness centers, concluded that users reflect more positive and meaningful perceptions of technician-led activities. In addition, the range of physical activity programs available is also an important factor influencing customer satisfaction. Clients are more likely to be satisfied if they can access a range of programs that meet their individual needs and preferences. For example, some people may prefer high-intensity training programs, while others may prefer lower-impact activities such as yoga or Pilates [24].

Therefore, after analyzing the previous literature that analyzes the variables of this study as the value chain of fitness centers and sports frequency, this research tries to extend this literature by analyzing these relationships through two different methodologies. Therefore, based on the analyzed literature, the following hypotheses are developed:

**Hypothesis** **1.***Within the hierarchical regression, sports frequency does not have a significant value in any of the models*.

**Hypothesis** **2.***Management variables are more explanatory in the linear models*.

**Hypothesis** **3.***In the Qualitative Comparative Analysis (QCA), in the 3 models analyzed, none of the conditions are necessary*.

**Hypothesis** **4.***With QCA, combinations appear that do not occur in the hierarchical regression*.

**Hypothesis** **5.***Sports frequency has a greater significant weight in the QCA analysis*.

## 3. Materials and Methods

### 3.1. Participants

The sample consisted of 383 users of a private sports facility with a mean age of 35.9 years (±15.79). The sample comprised a total of 201 men (52%) and 182 women (48%). This sports center offers a comprehensive range of sports facilities, including a weight training room, indoor swimming pool, and multi-purpose rooms for supervised activities. In terms of weekly sports practice, nine users (2.3%) go to the sports facility once a week, fifty-five users (14.3%) practice physical activity twice a week, one hundred and fifty-eight users go three times a week (41.2%), and one hundred and sixty-one users (42.2%) practice physical activity four times or more in the sports center.

### 3.2. Instrument

In order to know and be able to analyze the perceptions of the users of the sports service, they were surveyed by means of a questionnaire with a total of 49 indicators, made up of the following factors:

Perceived quality of service, a scale by Ko and Pastore [25] composed of 36 indicators, the response to which was submitted to a 5-point Likert scale (1 means strongly disagree and 5 means strongly agree). These 36 items were divided into 4 dimensions: quality of the program, quality of the interaction, quality of the outcome, and quality of the environment. The psychometric properties of this scale were confirmed in previous research [27].

User satisfaction, a scale by Hightower et al. [28] consisting of 2 indicators, the response to which was subject to a 5-point Likert scale (1 means strongly disagree and 5 means strongly agree). Both indicators measure the degree of user satisfaction with the sports service, a scale whose psychometric properties have been confirmed in previous studies [29].

User perceived value, a scale by Sweeney and Soutar [30] composed of 7 indicators, with a 5-point Likert scale response format (1 means strongly disagree and 5 means strongly agree). This scale was divided into 3 dimensions, price value, social value, and emotional value. The psychometric properties of this scale were confirmed in previous research [31].

Users’ future intentions, a scale by Zeithaml et al. [32] consisting of 4 indicators, with a 5-point Likert scale response (1 means strongly disagree and 5 means strongly agree). The psychometric properties of this scale were confirmed in recent studies [33].

Apart from these management variables, socio-demographic variables such as the sports frequency of the users and whether they attend directed activities or not were also included in the measurement instrument. Table 1 below shows the descriptive statistics of the variables analysed.

### 3.3. Procedure

The questionnaires were collected during the third quarter of 2019. During that time, and at the entrance of this sports facility, different collection times were scheduled in order to obtain the perceptions of a representative part of the users of the service. Purposive convenience sampling was used. Users were guaranteed that the data obtained would be anonymous and confidential, and they were willing to participate by relaying their perceptions through the measuring instrument. The research was conducted at the University of Valencia, where the requirement for ethical approval was not necessary. The Ethics and Human Research Committee at this university holds the belief that consent is not required to conduct a survey on a professional situation or topic with various perspectives.

### 3.4. Data Analysis

Once the questionnaires were collected, as shown in Table 2, the internal consistency of the scales used in the questionnaire was first checked with the data obtained, resulting in good values of Cronbach’s alpha (α), located above the point (0.70) required by the recommended literature [34].

The SPSS program (Statistical Package for the Social Sciences v.25) was used to obtain the descriptive data. Then, with the same statistical program, the hierarchical regression (HRM) of the analyzed variables was calculated, in order to see the evolution of the service management model through different steps, including sports frequency and attendance of supervised classes.

Finally, the analysis was performed with fsQCA, a method based on set theory. Through this analysis, it allows us to know the level that a variable must meet to influence a specific outcome when combined with other variables. There are three solutions for this analysis methodology: complex, parsimonious, and intermediate. This intermediate solution is the one used in this work, as suggested in the literature [35]. First, the calibration values of the data were established. Then, the set conditions are analyzed to finally perform a necessity and sufficiency analysis of the analyzed variables.

## 4. Results

### 4.1. Hierarchical Regression Model

Firstly, with regard to the hierarchical regression, it is observed that the models obtained predict between 52% and 58% of the variables analyzed, with the management variables offering the greatest predictive weight in the different models. As can be seen in Table 3, in the prediction of SAT, the management variables (SQ β = 0.58; *p* < 0.01) and PV (β = 0.17; *p* < 0.01)) gave a ΔR^2^ = 0.51 (*p* < 0.001), while the sports frequency variable (β = −0.04; *p* < 0.295) and class attendance (β = −0.02; *p* < 0.546) presented a ΔR^2^ = 0.01, concluding that the variables of both steps explain 52% of the satisfaction of the users of the sports service.

Regarding the VP model, the predictor variable service quality (SQ β = 0.75; *p* < 0.01) gave a ΔR^2^ = 0.56 (*p* < 0.001), while step 2, which adds sports frequency (β = −0.01; *p* < 0.171) and class attendance (β = −0.01; *p* < 0.124), presented a ΔR^2^ = 0.01, concluding that the variables of both steps explain 56% of the value perceived by users.

Finally, the different management variables together with the frequency and activity variables explain 58% of the future intentions of the users of the sports center. In this table, it is also observed in step 1 how the different management variables (CAL, SAT, and VP) obtain a predictive weight of 58% on future intentions (R^2^ = 0.58, *p* < 0.001). However, in step 2, when adding the variables of sports frequency and attendance to supervised classes, the prediction of the model improves in a very reduced way (R^2^ = 0.01, *p* < 0.05).

### 4.2. Qualitative Comparative Analysis

The descriptive statistics of the different dimensions and the calibration values are detailed below. Table 4 shows the calibration of the variables Service Quality (SQ), User Satisfaction (SAT), Perceived Value (PV), Sports Frequency (SF), and Attendance to Directed Classes (AD), as well as the values of the means and the deviation. The three thresholds obtained after recalibrating the values of the variables (10, 50, and 90) are also observed [36].

Subsequently, after the calibration of the variables, a necessity analysis is carried out to determine whether any of the existing conditions are necessary for the presence or absence of the results of the different dimensions that make up the management models of the sports centers. According to the literature [37], a condition is considered necessary when its consistency is greater than 0.90. Table 5 shows that none of the conditions are necessary for the presence or absence of any of the conditions analyzed, as they do not exceed the cut-off point suggested by the literature.

Finally, the sufficiency analysis of the conditions analyzed was carried out. Eng and Woodside [38] argue in their paper that models are adequate and informative when their consistency values are above 0.75. As can be seen in Table 6, in the different models, both for high and low levels of satisfaction (0.86; 0.76), perceived value (0.79; 0.84) and future intentions (0.89; 0.83) are above this cut-off point suggested by the literature, therefore, these values corroborate that they are informative and adequate models.

The combination of conditions that best explains the high levels of user satisfaction would be high levels of perceived quality*perceived user value (consistency: 0.92; raw coverage: 0.58). On the other hand, the configuration of combinations that best explains high levels of user-perceived value of the sports service is high levels of perceived quality of the service* low levels of attendance at guided activities (consistency: 0.78; raw coverage: 0.51). Finally, with regard to the positive results of the models, in order to obtain high levels of users’ future intentions, the combination of variables that best explains this is high levels of satisfaction*user perceived value (consistency: 0.90; raw coverage: 0.66). Thus, for positive future user behavior, the combination of user satisfaction and perceived value explains 66% of the variance.

At the same time, it is also important to know the combinations that imply low levels or not-so-positive user perceptions of the different models analyzed. First, the combination of conditions that most explains low levels of user satisfaction is low levels of perceived quality*low levels of perceived user value (consistency: 0.77; raw coverage: 0.79). As for the combination that would best explain low levels of perceived value, it would be low levels of perceived quality*low levels of sports frequency (consistency: 0.90; raw coverage: 0.30). Finally, for low levels of users’ future intentions towards the service, the combination of conditions that best explains it is low levels of satisfaction*low levels of perceived value by users (consistency: 0.85; raw coverage: 0.67). Looking at this last combination result, it explains that 67% of negative user behavior in the future will occur if there are low levels of satisfaction and low levels of perceived value.

## 5. Discussion

Fitness has undergone a very rapid and complex maturational evolution, starting as a market with slight acceptance where physical activity programs within sports services were very residual, or where it was not practiced on a regular basis. In contrast, the fitness market is currently valued as the main promoter of physical activity and healthy habits in our society [39,40].

The aim of this study was to find out the relationship that can be established between the variables that define the traditional management models of sports centers and the frequency of participation and attendance at supervised activities by their users. We also wanted to know what capacity and influence they have in predicting the future behavior of users in order to be able to offer meaningful information to the managers of these sports services. To this end, two different but complementary statistical methods were used: linear models and qualitative comparative analysis (QCA).

When looking at the linear models analyzed, it can be seen that the variables of the sports center management models have a greater predictive weight than the sport frequency or the attendance to supervised classes by the users of these sports services.

In this paper, within the linear models, perceived value is observed as the one that offers the greatest weight in the prediction of user loyalty and this predictive relationship has also been proven in other studies within the sports management literature [41,42]. However, the independent variables of frequency of sport participation and attendance to supervised classes have a very low predictive weight in the established models.

After these results were obtained from the linear method, where the contribution of each variable to the model can be seen in a biased way, an analysis of the models is carried out through fsQCA, which allows us to see the combinations of variables in the prediction of a model. With this methodology, it is possible to observe the relationships between the variables of the sports center management models, obtaining rational combinations of the conditions analyzed [43].

Management variables form the fundamental pillars in the prediction of the future intentions of sports center users, and this has been demonstrated in different studies found in the literature [33,44,45]. The results obtained with fsQCA when including frequency of participation and class attendance, unlike linear models, show that they have a significant influence when combined with management variables. Thus, for example, the results obtained show that for customers to have high levels of positive future behavior towards the sports service, they must be satisfied, perceive a positive value of the service, and have high levels of frequency of participation, as well as high attendance to fitness-guided classes.

From a sports management point of view, understanding the relationship between the frequency of sports participation and future intentions can help to develop strategies to increase participation and retention. By offering a wide range of sports activities as well as high-quality facilities and programs, sports centers can encourage users to participate more frequently and improve their intentions to continue using the center in the future.

Eime et al. [44] showed that individuals who participate in sports activities more frequently tend to have a greater sense of commitment and motivation towards their chosen sport. This may lead to a higher likelihood of continued participation in the future, as well as a higher likelihood of recommending the sports center to others.

In addition, sports centers that offer a wide range of sporting activities, as well as opportunities for socializing and community building, tend to have higher levels of customer satisfaction [45]. This is because people are more likely to find activities that match their interests and goals, and also have the opportunity to network with others who share similar interests. Providing quality guided activities is essential for the loyalty of users of these sports services, which requires qualified staff, promotion of the activities, and trying to create a sense of community. On the other hand, customers who experience poor service, lack of access to facilities, or a poor program may be less satisfied and therefore more likely not to repeat the sports service. Therefore, sports center management must be attentive to customer needs and complaints in order to keep customer satisfaction levels high. For example, Baena et al. [8] conclude that, in order to increase customer loyalty, it is important to offer a personalized and approachable service experience combined with service convenience, regardless of whether it is an in-person or virtual fitness class.

## 6. Conclusions

It is important for fitness centers to reinforce their activity programs and offer good quality facilities, thus trying to encourage a higher level of participation of the users of these sports services, and consequently reinforce positive behaviors in the future towards this service. Analyzing the sports frequency of the customers of a fitness center allows its managers to obtain significant information that reinforces the value chain and consequently the loyalty of these users. Using two different but complementary methodologies allows researchers to offer a wider range of results with more information and which helps them to understand the relationships between different user behaviors. This provides the managers of these sports services with valuable information that allows them to establish precise actions and strengthen the viability of the service.

## 7. Limitations, Future Lines of Research, and Practical Implications

Analyzing the frequency of attendance at sports centers is an important task for managers of sports centers to understand the current situation of physical activity in a given population. However, this research also has limitations and future research needs to be considered.

One of the most obvious limitations is the cause of the frequency of attendance at sports centers, such as being able to know various factors that may influence the attendance to these sports facilities, such as the location of the sports center, the offering of classes, the competition with other sports activities, and the availability of trained teachers. Another limitation encountered is that a non-probability sample was used, which limits the scope of representation of the whole population. However, from the observations made of the sample analyzed, it is considered that there is a rough estimate of the subject under investigation.

For these reasons, it is important to carry out future research to improve the accuracy and representativeness of the results obtained. Such research could include obtaining more detailed information on the frequency of attendance at sports centers through monitoring systems, as well as the consideration of demographic and economic factors in the analysis of the frequency of attendance. In this research, we have observed the important significance of perceived value on future intentions, as future research could analyze which dimension of this variable has the most influence on these future behaviors of sport service customers.

In addition, it would be useful to investigate the relationship between frequency of attendance and other important factors such as quality of life, physical and mental health, and prevention of chronic diseases.

This research yields results that can help to establish practical and managerial implications for improving the management of these sports services. From the point of view of the manager of these sports facilities, being able to analyze the sport frequentation of their users can be important to be able to detect trends and their needs, since their influence on user satisfaction can predetermine short-term decisions regarding their viability. Based on these needs, these professionals can design programs that respond to them, and expand their offerings by retaining users already enrolled, or even attracting new users.

Likewise, the analysis of the sports frequency of their users allows the managers of these sports services to outline their service strategies, since these sports patterns will help them to identify the hours of greatest affluence and, consequently, they will be able to define with greater precision the schedules and human resources to be employed at those times.

To promote high customer frequency and loyalty, sports facilities must offer a high-quality experience to their customers. This includes offering a wide range of sports and fitness activities that meet the needs and interests of a broad range of consumers, providing convenient and affordable access to facilities, and offering a supportive and welcoming environment. In addition, sports facilities can offer incentives, such as loyalty programs and special promotions, to encourage consumers to continue to participate in their activities and to recommend the facilities to others. Thus, the use of two complementary methodologies provides more complete information and more concisely helps sports managers to plan for and manage satisfied and loyal customers.

## Figures and Tables

**Table 1 healthcare-11-01320-t001:** Descriptive statistics of the dimensions analyzed.

	N	M	SD	S	K
Service Quality	383	3.98	0.49	−0.91	3.39
Satisfaction	383	3.77	0.72	−0.69	−0.65
Perceived Value	383	3.56	0.65	−0.34	−0.57
Future Intentions	383	3.56	0.71	−0.37	−0.65

Note. Mean (M), standard deviation (SD), skewness value (S) and kurtosis value (K).

**Table 2 healthcare-11-01320-t002:** Reliability of the scales.

	N	α	IC 95% α
Service quality	36	0.87	0.85–0.89
Overall satisfaction	2	0.82	0.78–0.85
Perceived value	7	0.83	0.80–0.85
Future intentions	4	0.86	0.83–0.88

Note: α, Cronbach alpha; IC95% α = Alpha Cronbach Interval.

**Table 3 healthcare-11-01320-t003:** Hierarchical regression models of management variables, sports frequency, and guided activity.

Variables	Satisfaction		Perceived Value		Future Intentions	
	ΔR^2^	β	ΔR^2^	β	ΔR^2^	β
Step 1	0.51 ***		0.56 ***		0.58 ***	
Service Quality		0.58 ***		0.75 ***		-
Satisfaction		-		-		0.25 ***
Perceived value		0.17 ***		-		0.59 ***
Step 2	0.01		0.01		0.01 *	
Service Quality		0.58 ***		0.75 ***		-
Satisfaction		-		-		0.25 ***
Perceived value		0.17 ***		-		0.58 ***
Sports frequency		−0.04		0.01		0.04
Fitness activities		−0.02		0.01		0.08 *
	0.52 ***		0.56 ***		0.58 ***	

Note. ΔR^2^, R-square change; β, standardized beta. * *p* ≤ 0.05; *** *p* ≤ 0.001.

**Table 4 healthcare-11-01320-t004:** Descriptive statistics and calibration values.

		SQ	SAT	PV	FI	SF	GA
Mean		361,940.892	17.54	18,774.542	327.81	5980.5642	3110.2720
SD		362,997.071	5.7	20,976.668	197.82	4894.99	3649.48
Minimum		1.5	1	1	8	36.00	9.00
Maximum		1,562,500	25	78,125	625	15,625.00	15,625.00
Percentiles	10	46,210.20	9.00	1728.00	81.00	1116.00	404.00
	50	245,367.50	16.00	11,520.00	256.00	4096.00	1728.00
	90	886,936.25	25.00	50,000.00	625.00	15,625.00	7500.00

Note. SQ, Service quality; SAT, Satisfaction; PV, Perceived value; FI, Future intentions; SF, Sports Frequency; GA, Guided Activities.

**Table 5 healthcare-11-01320-t005:** Necessity analysis for satisfaction, perceived value, and future intention.

	SAT		~SAT		PV		~PV		FI		~FI	
	Cons	Cov	Cons	Cov	Cons	Cov	Cons	Cov	Cons	Cov	Cons	Cov
CAL	0.7124	0.8755	0.4300	0.4115	0.7851	0.7645	0.4331	0.5249	-	-	-	-
~CAL	0.5211	0.5401	0.8699	0.7019	0.5122	0.4206	0.8057	0.8235	-	-	-	-
SAT	-	-	-	-	-	-	-	-	0.8200	0.7647	0.5498	0.4652
~SAT	-	-	-	-	-	-	-	-	0.4265	0.5108	0.7219	0.7845
PV	0.6805	0.8588	0.4441	0.4364	-	-	-	-	0.7328	0.8624	0.3925	0.4191
~PV	0.5534	0.5611	0.8563	0.6761	-	-	-	-	0.5064	0.4788	0.8711	0.7473
SF	0.8448	0.6298	0.8620	0.5004	0.8753	0.5171	0.8369	0.6153	0.8607	0.5983	0.8355	0.5270
~SF	0.3298	0.7544	0.3623	0.6452	0.3488	0.6321	0.3432	0.7741	0.3197	0.6818	0.3632	0.7029
FA	0.3492	0.5607	0.3512	0.4392	0.3486	0.4435	0.3513	0.5564	0.3758	0.5628	0.3217	0.4371
~FA	0.6507	0.5629	0.6487	0.4370	0.6516	0.4465	0.6486	0.5534	0.6241	0.5035	0.6782	0.4964

Note. ~, absence of condition; Cons, consistency; Cov, coverage; SAT: Satisfaction; PV: Perceived value; FI: Future intentions; SF: Sports frequency; FA: Fitness Activities.

**Table 6 healthcare-11-01320-t006:** Main conditions of sufficiency analysis for satisfaction, perceived value, and future intentions (intermediate solution).

Frequency Cuttoff: 1;	SAT			~SAT			PV		~PV		FI			~FI		
	Consistency Cuttoff: 0.86			Consistency Cuttoff: 0.79			Consistency Cuttoff: 0.80		Consistency Cuttoff: 0.86		Consistency Cuttoff: 0.86			Consistency Cuttoff: 0.85		
	1	2	3	1	2	3	1	2	1	2	1	2	3	1	2	3
SQ	⬤	⬤		◯		◯	⬤	⬤	◯	◯	-	-	-	-	-	-
SAT	-	-	-	-	-	-	-	-	-	-	⬤			◯		◯
PV	⬤		⬤	◯	◯		-	-	-	-	⬤	⬤	⬤	◯	◯	
FI	-	-	-	-	-	-	-	-	-	-	-	-	-	-	-	-
SF					◯	◯		◯	◯			⬤	◯		◯	◯
FA		◯	◯		⬤	⬤	◯			◯		⬤	◯			◯
Raw Coverage	0.58	0.46	0.44	0.79	0.14	0.13	0.51	0.28	0.30	0.27	0.66	0.22	0.16	0.67	0.33	0.17
Unique Coverage	0.12	0.06	0.05	0.66	0.01	0.01	0.34	0.11	0.18	0.15	0.31	0.02	0.01	0.40	0.65	0.01
Consistency	0.92	0.90	0.85	0.77	0.81	0.80	0.78	0.81	0.90	0.81	0.90	0.90	0.89	0.85	0.83	0.86
Overall solution consistency			0.86			0.76		0.79		0.84			0.89			0.83
Overall solution coverage			0.76			0.80		0.62		0.45			0.69			0.74

Note. ~, absence (low levels) of condition; ⬤ = presence (high level) of conditions; ◯ = absence (low levels) of condition; all sufficient conditions have an adequate raw coverage between 0.25 y, 65; SAT: Satisfaction, PV: Perceived value, FI: Future intentions, SF: Sports frequency, FA: Fitness activities. Expected vector for satisfaction: 1.1.1.1 (0: absent; 1: present), expected vector for ~ satisfaction: 0.0.0.0 expected vector for perceived value: 1.1.1, expected vector for ~ perceived value: 0.0.0, expected vector for future intentions: 1.1.1.1, expected vector for ~ future intentions: 0.0.0.0, using the format of Fiss (2011).

## Data Availability

Not applicable.

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
