# Peer review of "Influence of Sports Participation on the Behaviors of Customers of Sports Services: Linear and Qualitative Comparative Analysis Models"

_healthcare, 2023, doi:10.3390/healthcare11091320_

Round 1

Reviewer 1 Report

Introduction was written without citations.

Literature review was written to just show summary of the previous studies and failed to show a theoretical framework for the current research. I found no practical and theoretical contributions as well as strong justifications of this study after I read the literature review. I recommend that the author(s) should make clear hypotheses based on the theoretical backgrounds, which could guide your analysis and results. 

It seems that the authors attempted to analyze all relevant data to find some significances to talk about rather than to use a framework to test hypotheses.  Please work on tables to make readers easily understand. For example, for hierarchal regressions using step 1 and step 2, the authors used some variables as independent variables and dependent variables at the same time in the same table. It is very difficult to understand as readers.  If you have a hypothesis, please stick to the hypothesis and test only it rather than showing all unnecessary results. 

The authors said the tested the moderation effects, but what they did was mediating tests using table 3. There was no theoretical background to test 10 mediating effects and found one significant effect to talk about, which is not good practice in academic paper. 

There were 36 items for perceived quality, which should have several dimensions. All of sudden, it seems that the authors ignore all dimensions and used an overall perceived quality for this study.  In fact, other concepts seemed the same. 

Although the topic is very interesting, I am not sure the current form is ready for the publication in this journal.   

Author Response

The considerations and replies to the editors are in the attached file.

Reviewer 2 Report

I’m very grateful for the opportunity to review this manuscript, and I hope that my comments will help the authors to improve the current state of the document.

Introduction

The introduction is well planned, it’s simple and adequate to start the central topic of study.

Theoretical framework

In lines 64-65 the sentence is incomplete. The authors speak of "perceptions", however, I consider it necessary to better detail what constructs these perceptions are (quality, value, experience, convenience, etc.) and cite some research as an example.

In lines 95-96 it would be interesting to know the positive evolution of the last years. Can you provide data from 2 previous years?

In lines 120-122 the authors refer to Masuki et al. But I don't understand that statement. Do the authors mean that there is no control in sports centers? Are they referring to a specific model of fitness center?

Materials and method

Although the sample is well described, do you have more sociodemographic information? For example, stay time, travel time, type of fee, etc.

In the instrument, include all the dimensions of the Ko and Pastore scale as you do with the value scale, and also cite a study where this scale has been used with good psychometric properties, just as you do with the satisfaction and intention scale. future.

Both the procedure and the data analysis performed are adequate.

Results

I consider that the results obtained are well exposed, are clear and are focused on the objective of the study. However, there are several details that require review by the authors: 1) Tables 2 and 3 need a note; 2) Table 5 in line 300 should be table 6.

Discussion

What most attracts my attention is that the perceived value is the variable with the greatest influence on future intention. There are various studies in the fitness center sector that conclude that satisfaction is the most influential variable in future intention. As a future line, it could include analyzing what type of value (price, social or emotional) has the greatest influence.

Author Response

(The authors gave the same response as above.)

Reviewer 3 Report

The study examines a current topic. It is particularly important to examine physical activity after the Covid-19 epidemic, which had a significant impact. I suggest that some thoughts on this are included in the introduction. Especially since the data collection was done in 2019, it would be important to clarify that this was done before the epidemic, so it has no impact on the results.

I miss the summary of results in the abstract.

I find the introduction too general. The research concept could be developed in more detail.

Validated questionnaires were used for the research, which I consider appropriate.
The research methodology is relevant.
The derivation of the results is logical. The conclusions are correct.

In addition to the results obtained with two methods used, I miss the descriptive statistical summary of the results obtained with the questionnaires used. This could be done in a table.

Table 1 should be moved to the description of the research process (chapter 3.3).

I suggest developing a chapters 'discussion' and 'Manageral implications'.

There are some typos in the text (e.g. in point 32 of the bibliography).

In summary, I recommend the publication of the manuscript with minor corrections.

Author Response

(The authors gave the same response as above.)

Round 2

Reviewer 1 Report

The manuscript was improved after the revisions based on the comments.